# Association between Triglyceride Glucose Index and Corrected QT Prolongation in Chinese Male Steelworkers

**DOI:** 10.3390/ijerph18084020

**Published:** 2021-04-12

**Authors:** Thung-Lip Lee, Chin-Feng Hsuan, Cheng-Ching Wu, Wei-Chin Hung, I-Ting Tsai, Ching-Ting Wei, Teng-Hung Yu, I-Cheng Lu, Fu-Mei Chung, Yau-Jiunn Lee, Yung-Chuan Lu

**Affiliations:** 1Division of Cardiology, Department of Internal Medicine, E-Da Hospital, Kaohsiung 82445, Taiwan; lip1969@hotmail.com (T.-L.L.); calvin.hsuan@msa.hinet.net (C.-F.H.); maxvic24@gmail.com (C.-C.W.); p7411906@ms19.hinet.net (W.-C.H.); tenghungy@gmail.com (T.-H.Y.); chungfumei@gmail.com (F.-M.C.); 2School of Medicine for International Students, College of Medicine, I-Shou University, Kaohsiung 82445, Taiwan; triplet0826@gmail.com; 3School of Medicine, College of Medicine, I-Shou University, Kaohsiung 82445, Taiwan; tsai.iting@gmail.com; 4Division of Cardiology, Department of Internal Medicine, E-Da Dachang Hospital, Kaohsiung 80794, Taiwan; 5Department of Emergency, E-Da Hospital, Kaohsiung 82445, Taiwan; 6Division of General Surgery, Department of Surgery, E-Da Hospital, Kaohsiung 82445, Taiwan; 7Department of Biomedical Engineering, I-Shou University, Kaohsiung 82445, Taiwan; 8Department of Electrical Engineering, I-Shou University, Kaohsiung 82445, Taiwan; 9Department of Occupational Medicine, E-Da Hospital, Kaohsiung 82445, Taiwan; ed102431@edah.org.tw; 10Lee’s Endocrinologic Clinic, Pingtung 90000, Taiwan; lee@leesclinic.org; 11Division of Endocrinology and Metabolism, Department of Internal Medicine, E-Da Hospital, Kaohsiung 82445, Taiwan

**Keywords:** triglyceride glucose index, corrected QT prolongation, inter-relationship, structural equation modeling, steelworker

## Abstract

*Objectives:* Increased triglyceride glucose (TyG) index appears to be linked to carotid and coronary atherosclerosis and calcifications and possesses an elevated future risk of developing cardiovascular disease. Corrected QT (QTc) interval prolongation is associated with ventricular arrhythmias and sudden cardiac death, and a high prevalence of prolonged QTc interval was previously reported in blue-collar workers. The purpose of this study was to find the possible causal inter-relationship between TyG index and QTc interval in a large population of Chinese male steelworkers. *Methods:* A total of 3189 male workers from two steel plants were enrolled. They responded to a cross-sectional questionnaire on basic attributes and lifestyle, including sleep patterns. All workers in the two plants underwent periodic health checkups, including twelve-lead electrocardiography. Structural equation modeling (SEM) was used to assess the direct and indirect effects of TyG index on QTc interval. *Results:* With increasing TyG index tertile, the male steelworkers had an increased QTc interval. Applying multivariate analysis, TyG index was associated independently with the odds of QTc prolongation (adjusted odds ratio = 2.73, 95% confidence interval = 1.39–5.24, *p* = 0.004). SEM revealed that TyG index, hypertension, obesity, lifestyle, white blood cell (WBC) count, and liver function had statistically significant direct effects on QTc interval. Furthermore, TyG index also had an indirect effect on QTc interval through hypertension, obesity, WBC count, and liver function. Moreover, lifestyle had an indirect effect on QTc interval through TyG index. The final model explained 14% of the variability in QTc interval. *Conclusions:* An increased TyG index was associated with QTc interval prolongation in this study, and SEM delineated possible causal pathways and inter-relationships of the risk factors contributing to the occurrence of QTc prolongation among Chinese male steelworkers.

## 1. Introduction

Hypertriglyceridemia possesses independent risk factor for the development of disorders of glucose metabolism [1]. Increased plasma triglyceride levels are strongly associated with raised glucose levels because of interactions between muscle, adipose tissue, and pancreatic β-cells function [2,3]. Previous studies demonstrated that plasma triglyceride is an independent risk factor for the occurrence of fatty liver [4] and type 2 diabetes mellitus [5]. Furthermore, it was demonstrated that both fasting glucose and triglyceride levels in the high to normal range may predict future risk of developing cardiovascular disease (CVD) [6,7]. The triglyceride glucose (TyG) index is the product of fasting plasma glucose and triglyceride levels and a novel marker for metabolic disorders. Studies indicated that TyG index is positively associated with the occurrence of metabolic syndrome [8], calcification of coronary artery [9], atherosclerosis of carotid artery [10], and cardiovascular events [11].

The QT interval represents the time for both ventricular myocardial depolarization and repolarization [12]. Abnormal ventricular repolarization is the usual cause of QT interval prolongation [13]. Malignant arrhythmias [14] and sudden cardiac death in patients with myocardial infarction [15] are often seen in prolongation of heart rate-corrected QT (QTc) interval. Furthermore, QTc prolongation was shown to be an independent risk factor for CVD and mortality in the general population [16]. Moreover, the prevalence of QTc interval prolongation is reported to be higher in blue-collar workers [17], possibly due to hypertension, being overweight or obese, and smoking. We hypothesized that increased TyG index may play an important role in the occurrence of prolonged QTc interval in this group. If a close association between TyG index and QTc interval could be identified, strategies targeting improvements both in high fasting glucose and triglycerides may ameliorate QTc interval prolongation in blue-collar workers.

Therefore, the aim of this study was to investigate the relationship between TyG index and QTc interval using a standard baseline 12-lead electrocardiogram (ECG) in a large population of Chinese male steelworkers. In addition, we further assessed the effects of TyG index, lifestyle factors, hypertension, obesity, white blood cell (WBC) count, and liver function on QTc interval using a structural equation model (SEM) in this cross-sectional cohort.

## 2. Methods

### 2.1. Study Participants

This cross-sectional study recruited 3189 workers from two steel plants in Taiwan who received annual health check-ups from January 1 to December 31, 2016. The study hospital was the designated periodic health screening provider of the plants. The exclusion criteria were as follows: (1) atrial fibrillation or flutter, atrioventricular blocks, and bundle-branch blocks; (2) valvular heart disease, myocardial infarction, and history of cardiac surgery; (3) use of any medicine known to alter QT interval such as tricyclic antidepressants and class I (e.g., quinidine, mexiletine, procainamide, and flecainide) and class III (e.g., vernakalant, dronedarone, and amiodarone) antiarrhythmic medications; (4) liver disease, renal disease, or cancer of any origin; (5) inflammatory diseases (including infection or sepsis); (6) endocrine disorders that may affect glucose metabolism, such as hypothyroidism and hyperthyroidism; and (7) participants with a family history of long QT syndrome. The above information was confirmed in individual interviews conducted by occupational health physicians. The ethics approval of this study was obtained from the Human Research Ethics Committee of Kaohsiung E-Da Hospital. Informed written consent was obtained from all subjects enrolled in this study.

### 2.2. Baseline Data Collection

The self-management questionnaire was distributed and collected on the day of the health check. The questionnaire was used to determine basic demographic characteristics and lifestyle information, including age, gender, sleep quality, type of work, health status, physical exercise, betel nut chewing, smoking habits, drinking, medication use, and medical history. The smoking status of the participants was classified as a never-smoker, a former smoker (quit smoking for at least one year), or a current smoker. Betel quid chewing and alcohol drinking status were classified as never having chewed betel quid or drunk alcohol, former betel quid chewer or drinker (quit chewing betel quid or drinking alcohol for at least one year), or current betel quid chewer or drinker. In this study, former and current betel quid chewers and drinkers were categorized as a single group [18]. Body mass index (BMI) was calculated as the participant’s weight in kilograms divided by the square of height in meters. Obese (BMI ≥ 27.0 kg/m^2^) was defined according to the Bureau of Health Promotion, Department of Health, Taiwan. Hypertension was defined as continuously elevated systolic blood pressure (SBP) (≥140 mmHg) or diastolic blood pressure (DBP) (≥90 mmHg), or both. Participants who received antihypertensive therapy were also defined as suffering from hypertension. Subjects who met the following three or more criteria were defined as having metabolic syndrome: (1) systolic blood pressure of 130 mmHg or greater or diastolic blood pressure of 85 mmHg or greater, (2) central obesity (waist circumference greater than 80 cm for women and 90 cm for men), (3) serum triglyceride level of 150 mg/dL or greater, (4) serum high-density lipoprotein cholesterol (HDL-C) of less than 40 mg/dL for men or less than 50 mg/dL for women, and (5) fasting glucose of 100 mg/dL or greater or a previous diagnosis of type 2 diabetes. Diagnosis of diabetes mellitus (DM) was based on a level of glycated hemoglobin (HbA1c) ≥6.5% (48 mmol/mol) or elevated fasting blood glucose ≥126 mg/dL (7.0 mmol/L) or two-hour blood glucose ≥200 mg/dL (11.1 mmol/L), based on the 2016 American Diabetes Association (ADA) Guidelines [19], or a history of treatment of DM. Chronic kidney disease (CKD) was defined as an estimated glomerular filtration rate (eGFR) of <60 mL/min per 1.73 square meters.

### 2.3. Laboratory Measurements

After fasting for at least 8 hours, a peripheral blood sample was taken from the worker’s antecubital vein. Complete blood count and serum uric acid, aspartate aminotransferase (AST), alanine aminotransferase (ALT), glucose, HbA1c, and lipid panel (including plasma total cholesterol, low-density lipoprotein cholesterol (LDL-C), high-density lipoprotein cholesterol (HDL-C), and triglycerides) were also analyzed during the health check, and were determined in all workers using standard commercial methods with a parallel, multichannel analyzer (Hitachi 7170A, Tokyo, Japan) as described in our previous reports [20,21]. Peripheral leukocyte count was analyzed with an automated cell counter (XE-2100 Hematology Alpha Transportation System, Sysmex Corporation, Kobe, Japan). The TyG index was calculated as ln(fasting triglycerides (mg/dL) × fasting plasma glucose (mg/dL)/2) [22].

### 2.4. Electrocardiography, QT, and QTc Interval Measurements

When a patient was enrolled in the study, twelve-lead electrocardiogram was recorded for analysis during the baseline examination using a standardized protocol. QT interval was defined as the time from the first deflection of the QRS complex to the end of the T wave. The end of the T wave was determined by extending a tangent from the steepest downslope of the T wave until it crossed the TP segment. The QT and RR intervals were averaged over three consecutive complexes in lead II in sinus rhythm. The QTc interval was calculated using Bazett’s formula (QTc = QT/√RR) [23,24]. The ECG tracings were first analyzed by two independent cardiologists and the senior supervising cardiologist, who were blinded to the participants’ demographics. Discrepancies between readings were resolved through direct comparison, and a decision was made by a competent cardiologist. If the T wave amplitude was too flat, such that the end of the wave could not be identified, or if differences in the QTc measurements between the two independent cardiologists were too great, such that they could not be resolved by the supervisor, the data were excluded from the study. The final QTc value was the average of QTc values calculated by the supervisor and the two blinded independent cardiologists. In addition, extremely rapid (>150 bpm) and extremely slow (<40 bpm) heart rate recordings were excluded to eliminate the influence of heart rate on QT measurements [25,26]. According to previous research and suggestions from an ad hoc group regarding the latest European regulatory guidelines, QTc prolongation was divided into three gender-specific categories in this study [27]. The cutoff points for women were ≤450 ms (normal), 451 to 470 ms (borderline), and >470 ms (prolonged), while the cutoff points for men were ≤430 ms (normal), 431 to 450 ms (borderline), and >450 ms (prolonged).

### 2.5. Statistical Analysis

The Kolmogorov–Smirnov test was used to evaluate the normality of the data. Continuous normally distributed variables were expressed as the mean ± SD, and non-normally distributed variables were expressed as the median (interquartile range). A one-way analysis of variance (ANOVA) was used to compare the statistical differences of the normally distributed variables before Tukey’s pairwise comparisons were performed. Before statistical testing, the serum levels of triglycerides and ALT were logarithmically transformed to achieve a normal distribution. Categorical variables were reported as frequency, percentage, or both, and the χ^2^ test was used for comparisons between groups. Using normal and borderline QTc intervals as the reference category of patients, multiple logistic regression analysis was used to evaluate whether these variables had independent associations with abnormal QTc intervals.

Pearson’s correlation analysis was used to evaluate the relationship between QTC interval and biochemical and clinical variables. All statistical tests were two-tailed tests, and *p*-values less than 0.05 were considered statistically significant. SPSS software was used for all data analysis (SPSS for Windows, version 21.0; SPSS Inc., Chicago, IL, USA).

In addition, we designed one SEM model to assess the direct effect of TyG index on QTc interval independently of the indirect effect of TyG index on QTc interval mediated by other variables. The basic theoretical form of SEM is shown in Figure 1. Briefly, QTc interval was set as the dependent variable, and TyG index was set as the independent variable. Hypertension (confluence of SBP and DBP), obesity (confluence of BMI and waist circumference), liver function (confluence of ALT and AST), lifestyle (confluence of drinking, smoking, and betel nut chewing habits), and WBC counts were used as mediator variables, which were related to each other and QTc interval. The purpose of this analysis was to observe whether TyG index (independent variable) had a direct or indirect effect (through mediator variables) on QTc interval (dependent variable). In other words, although one independent variable seemed to directly affect QTc interval, the path diagram analysis indicated that this relationship was caused by the action of other independent variables acting as intermediators between the first independent variable and the QTc interval. We used standard criteria to evaluate the statistical fit of the model to the data, including root mean square error of approximation (RMSEA) values less than 0.08, comparative fit index (CFI) values exceeding 0.90, and standardized root mean square residual (SRMSR) values less than 0.06. Because it is too sensitive when applied to large datasets, the *χ*^2^ exact fit test was not used. In addition, the maximum-likelihood method was used to estimate the fit of the model, and standardized path coefficients and their statistical significance were presented. All data were analyzed using SEM and the path diagram analyses were performed with IBM SPSS AMOS (version 24.0; Amos Development Corp., Meadville, PA, USA) software.

## 3. Results

Table 1 shows the main characteristics of the 3189 participants stratified by TyG index. The mean TyG index was 4.7, and the median TyG index was also 4.7 (interquartile range, 4.5 to 4.9). The patients were divided according to tertiles of TyG index as follows: first tertile (<4.547), n = 1060; second tertile (4.547 to 4.810), n = 1069; and third tertile (>4.810), n = 1060. More of the participants in the third tertile (high TyG index) were aged 40–50 years and 50–60 years, and were associated with higher rates of alcohol and betel quid use, former and current smoking, hardly ever engaging in physical exercise, central obesity, hypertension, diabetes mellitus, metabolic syndrome, and chronic kidney disease, as well as an increased Framingham 10-year risk score. In addition, fewer of the participants in the third tertile were aged 25–40 years and never-smokers.

The inter-reader reproducibility evaluation of the QT measurements showed that the reliability coefficient was 0.995, and the Pearson’s correlation coefficient was 0.995. Comparing inter-reader QT measurements by the paired *t*-test did not reach statistical significance (*p* = 0.35).

The participants’ laboratory and echocardiographic data are listed in Table 2. Participants in the third tertile group had a higher left ventricular mass index than those in the first tertile. In addition, participants in the third tertile had higher SBP, DBP, BMI, waist circumference, calcium, fasting glucose, HbA1c, total cholesterol, triglycerides, AST, ALT, uric acid, albumin, WBC count, neutrophil count, monocyte count, lymphocyte count, heart rate, QRS duration, and QTc interval than those in the first and second tertiles did. The participants in the third tertile had lower eGFR and QT interval than those in the first tertile did, and also a lower HDL-C level than those in the first and second tertiles did. There were no significant differences in sodium, potassium, creatinine, ejection fraction, and PR interval among the three groups.

We performed a multivariate logistic regression analysis to evaluate the effects of TyG index and several other risk factors for arrhythmia in the presence of an abnormal QTc interval among such affected male steel workers. An abnormal QTc interval was positively associated with age, central obesity, and TyG index (Table 3).

Pearson’s correlation analysis showed that QTc interval was positively correlated with smoking, age, waist circumference, BMI, SBP, DBP, AST, ALT, HbA1c, total cholesterol, LDL-C, TyG index, albumin, WBC count, neutrophil count, monocyte count, and lymphocyte count, and negatively correlated with HDL-C (Table 4).

SEM was used to analyze the direct and indirect effects of TyG index on QTc interval (Figure 1). Figure 2 shows the estimated measured variable path analysis with parameters and the statistical significance of individual paths. The estimated model proved that the model fits well with a CFI of 0.971, an RMSEA of 0.059, and an SRMSR of 0.038 (Figure 2). TyG index, hypertension, obesity, WBC count, lifestyle, and liver function had statistically significant positive direct effects on QTc interval. Furthermore, TyG index also indirectly affected QTc interval through hypertension (β = 0.163), obesity (β = 0.375), WBC count (β = 0.165), and liver function (β = 0.139). Moreover, lifestyle indirectly affected QTc interval through TyG index (β = 0.273). The model explained 14% of the variability in QTc interval (Figure 2).

## 4. Discussion

The current study investigated the association between TyG index and QTc interval in a large population-based sample of Chinese male steelworkers. There are three main findings in this study. First, with increasing TyG index tertile, the participants had increased QTc interval and left ventricular mass index, and higher WBC count, neutrophil count, monocyte count, and lymphocyte count. Second, a higher TyG index was independently related to QTc prolongation—after adjustment for conventional risk factors including age, smoking, drinking, lack of physical exercise, shift work, hypertension, diabetes mellitus, central obesity, calcium, and eGFR—in the multiple regression analysis. Third, the causal relationship between TyG index and QTc interval was confirmed by SEM analysis.

QTc interval prolongation is not uncommon in blue-collar workers [17]. A prolonged QTc interval is proven to be related to the risk of cardiac arrhythmias and sudden cardiac death. Although several hypotheses—including traditional and nontraditional risk factors for cardiovascular disease such as diabetes mellitus, hypertension, uremic toxins, involvement of cardiac ion channels (analogous to inherited forms), and the development of cardiac autonomic neuropathy (a form of carotid and myocardial atherosclerotic disease)—have been proposed [28], the exact pathophysiological mechanisms underlying QT interval prolongation have yet to be clarified, especially in blue-collar workers. TyG index is composed of triglyceride and glucose levels. Previous studies demonstrated that serum triglycerides are associated with inflammation [29] and may therefore affect coronary microvascular dysfunction, and thus could be associated with the risk of cardiac failure [30], as well as an increase in baseline QTc [31]. Elevated triglyceride levels were also shown to be related to diabetes and obesity, which are known risk factors for prolongation of QT interval [32,33]. Previous studies demonstrated that arrhythmias are related to proinflammatory cytokines such as interleukin 6 and C-reactive protein by the modulation of ion-channel function [34,35] and aggravation of the sympathetic system [34]. These studies are consistent with our findings that with increasing TyG index tertile, the participants had increased QTc interval and left ventricular mass index, and higher WBC count, neutrophil count, monocyte count, and lymphocyte count. This raises the possibility that TyG index may be associated with inflammation processes that play a role in QTc interval prolongation.

TyG index is reported to be the best index for discriminating individuals with insulin resistance, even compared with other lipid parameters and visceral adiposity indicators [36]. A Korean cohort [37] observed a four-fold higher risk of developing diabetes mellitus in individuals in the highest baseline quartile of TyG index, and previously reported that changes in TyG index over time altered the risk and incidence of diabetes mellitus [38]. TyG index was also proposed as a candidate marker for classifying metabolic health status [39]. In further evidence of its potential as a novel marker, TyG index is reported to have high sensitivity and specificity in identifying metabolic syndrome [8]. Several studies also demonstrated the relationship between TyG index and carotid atherosclerosis, coronary artery calcification, a high risk of cardiovascular disease, and accurate prediction of cardiovascular outcomes [11,40,41,42]. In blue-collar workers, age, hypertension, and body mass index were reported to be independent predictors of QT interval prolongation [17]. In our study, an increased TyG index had a relatively direct effect on QTc interval and was significantly associated with QTc interval prolongation, which represents ventricular myocardial membrane electrical stabilization, in male steelworkers.

The present study is the first of its kind to explore associations between TyG index and QTc interval in male steelworkers to date. However, the exact mechanism underlying the association between TyG index and QTc interval was not yet fully clarified. Our SEM analysis revealed that there were significant positive direct effects from TyG index, hypertension, obesity, WBC count, lifestyle, and liver function on QTc interval. Furthermore, TyG index also had an indirect effect on QTc interval through hypertension, obesity, WBC count, and liver function. Moreover, lifestyle had an indirect effect on QTc interval through TyG index. Previous studies demonstrated associations between hypertension [43], obesity [32], inflammation [44], lifestyle [45], and liver disease [46] and QTc prolongation. TyG index is reported to be a simple surrogate of insulin resistance [47], and numerous studies also found a relationship between insulin resistance and hypertension [48,49]. In an animal study, Lin et al. [50] found that insulin resistant, obese, 16 to 17-week-old rats exhibited defective inactivation of current, altered electrophysiology characteristics, and developed cardiac hypertrophy due to a prolongation of QTc interval. Notably, inflammatory markers inducing atherosclerotic plaque instability, such as interleukins, tumor necrosis factor-α, fibrinogen, and leukocytes, were shown to play crucial roles in QTc prolongation and related disorders [44]. Jin et al. [42] also found a relationship between inflammation and TyG index. Furthermore, prolonged QTc interval was observed both in patients with liver cirrhosis and in those with nonalcoholic fatty liver disease (NAFLD). Bellan et al. [51] suggested that, independent of the etiology of chronic liver disease, liver fibrosis is a stronger determinant of prolonged QTc interval than fatty liver. In addition, TyG index was positively related to the presence of liver fibrosis and the severity of hepatic steatosis in NAFLD [52]. Hence, it is reasonable to suggest that TyG index may be associated with hypertension, obesity, WBC count, and liver function, thereby contributing to QTc interval prolongation.

In addition, prolonged QTc interval may be associated with cigarette smoking [53] and alcohol consumption [54]. Several studies demonstrated that smoking is a major cause of atherosclerosis-related cardiovascular events. Moreover, smoking may lead to ventricular arrhythmias and sudden cardiac death [55]. A previous study reported that alcohol can inhibit Na-K-ATPase activity [56], and a decrease in Na-K-ATPase pump activity was reported to alter resting membrane potential across the intracellular and extracellular ionic and sarcolemma homeostasis [57]. Furthermore, during action potential plateau, drinking may affect the number of calcium ions entering cardiac cells by voltage-dependent calcium channels, and may affect the activity of voltage-dependent calcium channels in the sarcolemma [58]. Therefore, ventricular repolarization, which depends on an increase in outward K currents and a reduction in L-type calcium currents, may be prolonged through alcohol [59]. Moreover, betel quid chewing is associated with myocardial infarction, cardiac arrhythmias, and central obesity [60]. Therefore, chewing betel quid may lead to QTc interval prolongation. Additionally, chewing betel quid, smoking tobacco, and consuming alcohol are associated with metabolic syndrome, hypertriglyceridemia, and insulin resistance [61,62,63,64]. Therefore, it is reasonable to suggest that lifestyle has an indirect effect on QTc interval through TyG index. Taken together, these findings suggest that changes to these modifiable risk factors can provide significant long-term benefits to male steelworkers.

There are several limitations to this study. First, the cross-sectional design limits our ability to infer a causal relationship between increased TyG index and QTc prolongation. Studies with long-term follow-up are needed to clarify the association between TyG index and QTc prolongation. Second, our study was performed in a Chinese population, and these findings may lack generalizability to other populations. Third, the heart rate and QT interval of the study participants were measured by the same medical technician using an identical method. A limitation of the present study is that the QT intervals were measured by a computer-based method and that variability of the QT interval was not assessed. Fourth, we could not analyze our data stratified by sex because of the small number of female steelworkers. Fifth, in the present study, there were no data of postprandial triglyceride and glucose, which might affect QTc prolongation.

## 5. Conclusions

We indicated that increased TyG index was associated with QTc prolongation on standard baseline 12-lead ECG. In addition, SEM delineated inter-relationships of the risk factors and the potential pathways that may contribute to the development of QTc prolongation in Chinese male steelworkers.

## Figures and Tables

**Figure 1 ijerph-18-04020-f001:**
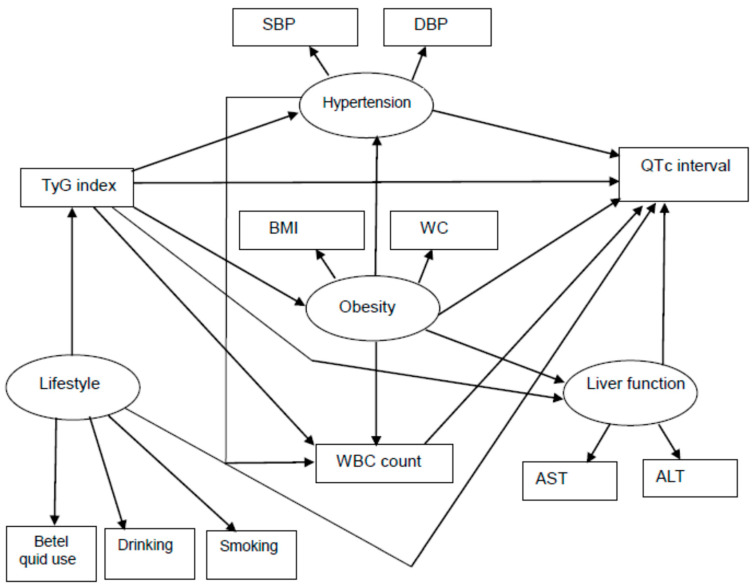
Structural equation models for corrected QT interval; ellipses represent latent variables; boxes represent observed variables; straight lines with one arrowhead indicate direct effects. Abbreviations: SBP, systolic blood pressure; DBP, diastolic blood pressure; TyG, triglyceride glucose; QTc, corrected QT; BMI, body mass index; WC, waist circumference; WBC, white blood cell; AST, aspartate aminotransferase; ALT, alanine aminotransferase.

**Figure 2 ijerph-18-04020-f002:**
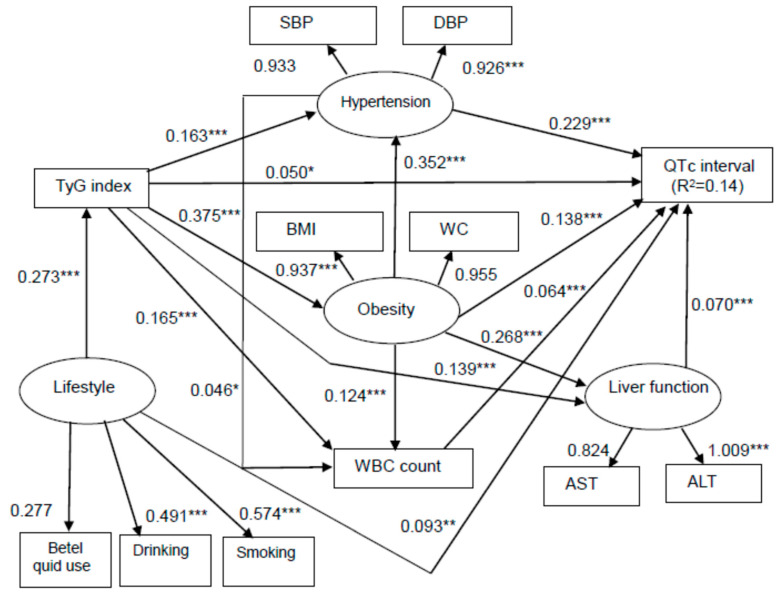
Structural equation model for corrected QT interval in the male steelworkers. Comparative fit index (CFI), 0.971; goodness of fit index (GFI), 0.974; root mean square error of approximation (RMSEA), 0.059; standardized root mean square residual (SRMSR), 0.038. * *p* < 0.05, ** *p* < 0.01, and *** *p* < 0.001. Path loadings are standardized coefficients. SBP, systolic blood pressure; DBP, diastolic blood pressure; TyG, triglyceride glucose; QTc, corrected QT; BMI, body mass index; WC, waist circumference; WBC, white blood cell; AST, aspartate aminotransferase; ALT, alanine aminotransferase.

**Table 1 ijerph-18-04020-t001:** Main characteristics according to tertile of triglyceride glucose index.

Variable	First Tertile <4.547	Second Tertile 4.547–4.810	Third Tertile >4.810	*p*-Value
No.	1060	1069	1060	
Age (years) (n, %)				
25–40	544 (51.3)	417 (39.0)	320 (30.2)	<0.0001
40–50	376 (35.5)	484 (45.3)	547 (51.6)	<0.0001
50–60	123 (11.6)	151 (14.1)	173 (16.3)	0.007
>60	17 (1.6)	17 (1.6)	20 (1.9)	0.836
Alcohol use (n, %)	263 (24.8)	304 (28.4)	387 (36.5)	<0.0001
Betel quid use (n, %)	7 (0.7)	9 (0.8)	33 (3.1)	<0.0001
Smoking (n, %)				
Never	575 (54.3)	514 (48.1)	402 (37.9)	<0.0001
Former	89 (8.4)	112 (10.5)	128 (12.1)	0.020
Current	314 (29.6)	374 (35.0)	476 (44.9)	<0.0001
Physical exercise in the past month (n, %)				
Hardly ever	249 (23.5)	258 (24.1)	315 (29.7)	0.003
Once	612 (57.7)	618 (57.8)	581 (54.8)	0.313
Twice or more	199 (18.8)	193 (18.1)	164 (15.5)	0.132
Poor sleep (n, %)				
Almost never	802 (75.7)	817 (76.4)	785 (74.1)	0.474
Sometimes	182 (17.2)	171 (16.0)	191 (18.0)	0.480
Often or almost always	76 (7.2)	81 (7.6)	84 (7.9)	0.835
Central obesity (n, %)	123 (11.6)	248 (23.2)	381 (35.9)	<0.0001
Hypertension (n, %)	255 (24.1)	398 (37.2)	519 (49.0)	<0.0001
Diabetes mellitus (n, %)	218 (20.6)	366 (34.2)	587 (55.4)	<0.0001
Metabolic syndrome (n, %)	27 (2.6)	124 (11.6)	587 (55.4)	<0.0001
Chronic kidney disease (n, %)	16 (1.5)	41 (3.8)	52 (4.9)	<0.0001
Shift work (n, %)	473 (44.6)	455 (42.6)	475 (44.8)	0.573
Framingham 10-year risk score (median, interquartile range)	1.3 (0.5–2.7)	2.5 (1.2–4.8)	4.2 (2.1–7.4)	<0.0001

**Table 2 ijerph-18-04020-t002:** Laboratory and echocardiographic data according to tertiles of triglyceride glucose index.

Variable	First Tertile <4.547	Second Tertile 4.547–4.810	Third Tertile >4.810	*p*-Value
No.	1060	1069	1060	
Systolic blood pressure (mmHg)	119 ± 14	124 ± 15	128 ± 16	<0.0001
Diastolic blood pressure (mmHg)	75 ± 9	79 ± 11	82 ± 11	<0.0001
Body mass index (kg/m^2^)	23.6 ± 3.2	25.1 ± 3.3	26.3 ± 3.6	<0.0001
Waist circumference (cm)	80.2 ± 8.3	84.4 ± 8.1	87.3 ± 8.5	<0.0001
Sodium (mEq/L)	140.3 ± 1.6	140.3 ± 1.6	140.2 ± 1.7	0.051
Potassium (mEq/L)	4.02 ± 0.28	4.02 ± 0.27	4.02 ± 0.29	0.865
Calcium (mg/dL)	9.5 ± 0.3	9.6 ± 0.3	9.6 ± 0.4	<0.0001
Fasting glucose (mg/dL)	94.4 ± 9.6	99.0 ± 13.2	109.8 ± 30.6	<0.0001
HbA1c (%)	5.5 ± 0.4	5.6 ± 0.5	5.9 ± 1.0	<0.0001
Total cholesterol (mg/dL)	181.3 ± 31.1	193.7 ± 31.3	205.7 ± 35.6	<0.0001
Triglyceride (mg/dL)	69.5 (57.0–81.0)	116.0 (102.0–133.0)	208.0 (170.0–275.8)	<0.0001
HDL cholesterol (mg/dL)	51.6 ± 10.8	46.1 ± 9.1	41.8 ± 7.8	<0.0001
LDL cholesterol (mg/dL)	104.2 ± 27.6	117.0 ± 27.8	116.0 ± 31.7	<0.0001
Aspartate aminotransferase (U/L)	26.8 ± 9.0	30.9 ± 17.1	34.5 ± 18.2	<0.0001
Alanine aminotransferase (U/L)	27.0 (20.0–37.0)	33.0 (25.0–48.0)	40.0 (30.0–59.0)	<0.0001
Uric acid (mg/dL)	6.2 ± 1.2	6.6 ± 1.3	6.9 ± 1.4	<0.0001
Creatinine (mg/dL)	1.15 ± 0.17	1.17 ± 0.27	1.18 ± 0.47	0.149
Albumin (g/dL)	4.4 ± 0.2	4.5 ± 0.2	4.5 ± 0.2	<0.0001
Estimated GFR (ml/min/1.73 m^2^)	79.6 ± 10.5	77.5 ± 10.9	77.0 ± 11.1	<0.0001
White blood cell count (×10^9^/L)	5.874 ± 1.528	6.296 ± 1.557	6.747 ± 1.646	<0.0001
Neutrophil count (×10^9^/L)	3492 ± 1239	3698 ± 1189	3930 ± 1238	<0.0001
Monocyte count (×10^9^/L)	333 ± 118	352 ± 116	371 ± 119	<0.0001
Lymphocyte count (×10^9^/L)	1863 ± 525	2042 ± 564	2226 ± 618	<0.0001
Ejection fraction (%)	68.3 ± 5.4	68.3 ± 5.8	68.7 ± 5.5	0.366
Left ventricular mass index (g/m^2^)	91.5 ± 17.8	92.9 ± 17.6	95.0 ± 19.4	0.016
ECG parameters				
Heart rate (bpm)	64.4 ± 9.5	66.1 ± 9.2	68.8 ± 9.8	<0.0001
PR interval (ms)	158.7 ± 39.5	158.1 ± 19.5	159.0 ± 20.5	0.764
QRS duration (ms)	93.9 ± 10.6	93.8 ± 13.1	95.2 ± 11.0	0.012
QT interval (ms)	395.4 ± 25.8	393.0 ± 24.5	390.7 ± 23.9	0.0002
QTc interval (ms)	406.3 ± 21.1	408.8 ± 22.8	415.2 ± 20.7	<0.0001

Data are mean ± SD or median (interquartile range). HbA1c, glycated hemoglobin; HDL, high-density lipoprotein; LDL, low-density lipoprotein; GFR, glomerular filtration rate; QTc, corrected QT.

**Table 3 ijerph-18-04020-t003:** Multivariate logistic regression analysis with the presence of abnormal corrected QT interval as the dependent variable.

Variable	exp(B)	95% Confidence Interval	*p*-Value
Age	1.05	1.01–1.09	0.006
Smoking	0.95	0.58–1.52	0.821
Alcohol use	0.68	0.40–1.13	0.141
Lack of physical exercise	1.46	0.89–2.34	0.128
Shift work	1.19	0.76–1.86	0.440
Hypertension	1.82	0.99–3.22	0.055
Diabetes mellitus	1.16	0.37–3.00	0.787
Central obesity	1.80	1.11–2.88	0.018
Calcium	0.99	0.55–1.86	0.975
Estimated GFR	0.10	0.98–1.02	0.957
TyG index	2.73	1.39–5.24	0.004

GFR, glomerular filtration rate; TyG, triglyceride glucose.

**Table 4 ijerph-18-04020-t004:** Correlation matrix between QTc interval and clinical and biochemical measures.

Variable	Age	Smoking	Drinking	Betel Use	Shift Work	BMI	WC	SBP	DBP	AST	ALT	HbA1c	T-CHOL	HDL-C	LDL-C	TyG	Cr	eGFR	Albumin	WBC	Monocyte	Neutrophil	Lymphocyte	QTc
Age	1	−0.04 *	0.03	−0.01	−0.07 **	0.00	0.04 *	0.11 **	0.17 **	0.04 *	−0.04 *	0.24 *^*^	0.11 **	−0.05 **	0.07 **	0.17 **	0.03	−0.37 **	−0.25 **	−0.04 *	−0.06 **	−0.02	−0.07 **	0.17 **
Smoking		1	0.28 **	0.15 **	0.05 **	0.00	0.02	−0.07 **	−0.06 **	0.01	0.03	0.05 **	−0.00	−0.10 **	−0.00	0.15 **	−0.03	0.05 **	−0.06 **	0.26 **	0.27 **	0.20 **	0.20 **	0.04 *
Drinking			1	0.13 **	−0.00	0.04 *	0.06 **	−0.00	0.03	0.02	−0.01	0.03	0.00	0.02	−0.04 *	0.11 **	−0.02	−0.00	−0.04 *	0.05 **	0.06 **	0.04 *	0.04 *	0.03
Betel use				1	−0.02	0.00	0.02	0.00	0.00	0.03	0.02	0.02	−0.01	−0.02	−0.06 **	0.11 **	0.00	−0.01	−0.02	0.04 *	0.04 *	0.02	0.06 **	0.01
Shift work					1	0.01	0.01	0.02	0.01	−0.02	0.01	−0.02	−0.00	−0.01	0.01	0.00	−0.01	0.04 *	−0.01	0.06 **	0.05 **	0.04 *	0.04 *	−0.00
BMI						1	0.89 **	0.39 **	0.35 **	0.23 **	0.31 **	0.18 **	0.12 **	−0.34 **	0.16 **	0.34 **	0.03	−0.03	−0.02	0.19 **	0.14 **	0.13 **	0.21 **	0.24 **
WC							1	0.37 **	0.35 **	0.24 **	0.31 **	0.20 **	0.14 **	−0.33 **	0.16 **	0.37 **	0.01	0.00	−0.03	0.19 **	0.15 **	0.12 **	0.21 **	0.29 **
SBP								1	0.86 **	0.17 **	0.16 **	0.15 **	0.08 **	−0.10 **	0.05 *	0.25 **	0.09 **	−0.09 **	0.10 **	0.13 **	0.05 **	0.13 **	0.08 **	0.29 **
DBP									1	0.18 **	0.17 **	0.16 **	0.14 **	−0.09 **	0.08 **	0.29 **	0.05 **	−0.12 **	0.08 **	0.12 **	0.05 **	0.12 **	0.07 **	0.29 **
AST										1	0.83 **	0.10 **	0.09 **	−0.08 **	0.05 **	0.22 **	0.01	−0.02	0.04 *	0.07 **	0.09 **	0.02	0.12 **	0.18 **
ALT											1	0.11 **	0.12 **	−0.17 **	0.12 **	0.25 **	−0.02	0.04 *	0.06 **	0.10 **	0.12 **	0.04 *	0.15 **	0.17 **
HbA1c												1	0.15 **	−0.13 **	0.11 **	0.35 **	−0.00	−0.03	−0.05 **	0.12 **	0.06 **	0.08 **	0.15 **	0.14 **
T-CHOL													1	0.14 **	0.82 **	0.35 **	−0.00	−0.12 **	0.11 **	0.07 **	0.00	0.01	0.15 **	0.08 **
HDL-C														1	−0.05 **	−0.44 **	−0.04 *	0.01	0.04 *	−0.14 **	−0.11 **	−0.09 **	−0.15 **	−0.11 **
LDL-C															1	0.12 **	−0.02	−0.07 **	0.09 **	0.09 **	0.04 *	0.04 *	0.14 **	0.06 **
TyG index																1	0.04 *	−0.10 **	0.07 **	0.22 **	0.13 **	0.14 **	0.26 **	0.18 **
Cr																	1	−0.55 **	−0.01	−0.02	0.00	−0.00	−0.05 **	0.02
eGFR																		1	0.05 *	0.05 **	0.04 *	0.04 *	0.05 **	−0.03
Albumin																			1	0.07 **	−0.01	0.07 **	0.07 **	0.04 *
WBC count																				1	0.67 **	0.91 **	0.63 **	0.14 **
Monocyte count																					1	0.55 **	0.43 **	0.08 **
Neutrophil count																						1	0.27 **	0.12 **
Lymphocyte count																							1	0.11 **
QTc interval																								1

BMI, body mass index; WC, waist circumference; SBP, systolic blood pressure; DBP, diastolic blood pressure; AST, aspartate aminotransferase; ALT, alanine aminotransferase; T-CHOL, total cholesterol; HDL-C, high-density lipoprotein cholesterol; LDL-C, low-density lipoprotein cholesterol; TyG, triglyceride glucose; Cr, creatinine; eGFR, estimated glomerular filtration rate; WBC, white blood cell; QTc, corrected QT. * *p* < 0.05; ** *p* < 0.01.

## Data Availability

The data presented in this study are available on request from the corresponding author.

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
