# Peer review of "Association between Triglyceride Glucose Index and Corrected QT Prolongation in Chinese Male Steelworkers"

_ijerph, 2021, doi:10.3390/ijerph18084020_

Round 1

Reviewer 1 Report

This study was a cross-sectional, observational study, which aimed to examine the relationship between TyG index population-based sample of Chinese male steelworkers and the effects of TyG index, lifestyle factors, hypertension, obesity, white blood cell count, and liver function on QTc interval using a structural equation model. The main findings were (1) with increasing TyG index tertile, the participants had increased QTc interval and left ventricular mass index, and higher WBC count, neutrophil count, monocyte count, and lymphocyte count, (2) a higher TyG index was independently associated with QTc prolongation after controlling for conventional risk factors including age, smoking, alcohol use, lack of physical exercise, shift work, hypertension, diabetes mellitus, central obesity, calcium, and eGFR in the multiple regression analysis, and (3) SEM analysis confirmed the causal relationship between TyG index and QTc interval. This reviewer considers that the present study is clinically important, and that the authors well performed the present study. This reviewer has some comments as described below.

Major comments:

  1. The present study was performed using fasting blood samples. Thus, there were no data of postprandial triglyceride and glucose, which might affect QTc prolongation. The authors should add this issue in the Limitation section.
  2. Some amounts of participants had comorbidities, including hypertension, diabetes, metabolic syndrome. Are there medical treatment data of the participants?

Reviewer 2 Report

This is a very interesting paper, which draws the readers attention. I have some suggestions to make to the authors.

1.What about the family history for long QT syndrome? Were participants with such family history excluded?

2. The part regarding inter reader reproducibility assessment results, should appear in the result section.

Reviewer 3 Report

In general,

Use italics for mellitus.

Homogenize the font in the writing of the article.

Homogenize the wording of the bibliographic references according to the guidelines for authors, section references, this because there are differences in the wording between them, for example, points between the abbreviated name, point between the abbreviated name of the reference bibliography and the year, that is, some are written in the way mentioned above and others are not, then homogenize the writing of the references.
